New craniodental remains of Wakaleo alcootaensis (Diprotodontia: Thylacoleonidae) a carnivorous marsupial from the late Miocene Alcoota Local Fauna of the Northern Territory, Australia

Yates Adam M. adamm.yates@nt.gov.au
Museum of Central Australia, Museum and Art Gallery of the Northern Territory , Alice Springs, Northern Territory , Australia
Piñeiro Graciela
Electronic publication date: 2015 Nov 12
Publication date: 2015
Volume: 3
Electronic Location ID: e1408
Received 2015 Jul 31; Accepted 2015 Oct 26
Copyright: © 2015 Yates
Copyright year: 2015
Copyright holder: Yates
License: This is an open access article distributed under the terms of the Creative Commons Attribution License, which permits unrestricted use, distribution, reproduction and adaptation in any medium and for any purpose provided that it is properly attributed. For attribution, the original author(s), title, publication source (PeerJ) and either DOI or URL of the article must be cited.
License URL: https://creativecommons.org/licenses/by/4.0/

Keywords: Miocene, Australia, Thylacoleonidae, Wakaleo, Alcoota, Parallel evolution

Funding: The author received no funding for this work.

==============================
New jaws and teeth referable to the rare thylacoleonid marsupial Wakaleo alcootaensis are figured and described. The species is the geologically youngest known member of the genus and is only known from the late Miocene Alcoota Local Fauna of the Northern Territory, Australia. A revised diagnosis of the species is presented which is found to be morphologically distinct from its congeners. W. alcootaensis can be distinguished from other species of Wakaleo by its greater size, deeply recessed masseteric fossa, more steeply angled I1, loss of P2, greater P3 to M1 ratio and loss of M3. Several characters of W. alcootaensis, including the increase in size, steeply angled I1, increase of the relative size of P3, and reduction of the molar row are present in at least some species of Thylacoleo. Phylogenetic analysis suggests that these character states are convergences and that there was parallel evolution in these two thylacoleonid lineages.

Introduction

Thylacoleonids, or ‘marsupial lions’ are a group of small to large-bodied diprotodontian marsupials that range from the size of small house cat in Priscileo roskellyae (Wroe et al., 2003) up to the size of a lion in Thylacoleo carnifex (Wroe et al., 1999; Wroe et al., 2003). They are characterised by the development of the third premolar pair into large shearing blades. Although there has been much debate about the diet of these creatures in the past, it is now largely accepted that Sir Richard Owen was correct in 1859 when he described the eponymous Thylacoleo carnifex as “one of the fellest and most destructive of predatory beasts” (Owen, 1859, pg. 319).

Wakaleo is a genus of thylacoleonids that ranges from the late Oligocene through to the late Miocene (Gillespie, 2007). It differs from the Plio-Pleistocene genus, Thylacoleo, by having highly reduced to absent anterior premolars, a P3 that broadens posteriorly, a cuspule-like anterolingual crest on the P3, less reduced molars, lack of a fronto-squamosal suture and no postorbital bar amongst other features (Murray, Wells & Plane, 1987; Gillespie et al., 2014).

The described species of Wakaleo form an apparent evolutionary sequence that matches their stratigraphic occurrence (Archer & Dawson, 1982; Murray & Megirian, 1990) and it has been suggested that the genus is one of the more usefull mammalian lineages for biochronolgy of Australian mammal-bearing deposits (Megirian et al., 2010; Arena et al., 2015).

The oldest of the described species currently placed in Wakaleo is the type species, W. oldfieldi Clemens & Plane, 1974 which was based on specimens from the Kutjamparpu Local Fauna of the Wipijiri Formation at Lake Ngapakaldi, South Australia. The Kutjamarpu Local Fauna has had various age estimates but it is most securely correlated with Faunal Zone B local faunas from the Riversleigh World Heritage Area which have recently produced radiometric dates of 16.5–18.2 ma, i.e., early Miocene (Woodhead et al., 2014). W. vanderleueri Clemens & Plane, 1974 is a younger and slightly larger species based on remains from the Bullock Creek Local Fauna of the Northern Territory. Apart from its larger size it can be distinguished from W. oldfieldi by its larger P3 to M1 ratio (Clemens & Plane, 1974) and the loss of the talonid shelf on M3 (Gillespie et al., 2014). The Bullock Creek Local Fauna has been consistently regarded as middle Miocene in age and correlated with Faunal Zone C local faunas of Riversleigh. Recent radiometric dating of Faunal Zone C sites has produced dates of 13.5–15.1 ma, confirming their middle Miocene age (Woodhead et al., 2014). Both of these species have recently been reported from the Riversleigh World Heritage area. Some of the W. oldfieldi fossils are from Faunal Zone B sites, as is to be expected, while the bulk of the W. vanderleueri specimens come from the even younger Faunal Zone D Encore Site, upholding the stratigraphic separation of the two species. However both species have been found in Riversleigh Faunal Zone C sites. Nevertheless they do not co-occur in the same local faunas. The biochronologically important diprotodontid genus, Neohelos, also displays a species turnover within Faunal Zone C, indicating that the zone spans a biochronologically significant length of time (Arena et al., 2015). Indeed, Arena et al. (2015) were able to divide System C site into three faunal intervals based on mammalian lineages. The sites bearing W. oldfieldi belong to the older two intervals while the only Faunal Zone C site that bears W. vanderleueri, Golden Steph Site, belongs to the youngest of the three intervals (Arena et al., 2015).

The youngest, and largest, known member of Wakaleo is W. alcootaensis Archer & Rich, 1982, from the late Miocene Alcoota Local Fauna of central Australia (Murray & Megirian, 1992; Fig. 1). This species has measurements that exceed W. vanderleueri by up to a third (Archer & Rich, 1982; this paper). Given that Wroe et al. (1999) estimated the size range of 44–56 kg for W. vanderleueri we can tentatively estimate a weight range of up to 96–123 kg for W. alcootaensis assuming geometric similitude between the two species. The Alcoota Local Fauna is dominated by large browsing herbivores, both mammalian and avian (Murray & Megirian, 1992), whereas mammalian carnivores are exceptionally rare. At approximately the size of a small lioness, W. alcootaensis was the largest of these. Unfortunately the species has remained extremely rare and poorly known. Indeed, anatomical knowledge of the species is so poor that it could only be diagnosed by its larger size relative to other species of Wakaleo, leaving open the question of its validity as a distinct taxon even though this question has not been raised in the literature. The species was established for a single cranial fragment that was unfortunately badly damaged while trenching around a plastered block of dense bone bed material (Archer & Rich, 1982). Very few other specimens of this species have been found. One of them is a dentary fragment bearing two molars (UCMP 65621) that was recovered during an initial investigation of Alcoota in 1962 (Fig. 2). This specimen was initially described as a possible giant perameloid (Woodburne, 1967). G Prideaux (pers. comm., 2012) first suggested that this specimen was actually a misidentified Wakaleo specimen, a reidentification that is supported in the present work. The only other positively attributable specimens that came to light prior to 2013 are a few isolated teeth and a few postcranial elements, none of which have been described in the scientific literature.

Figure 1 Map of north-west corner of Alcoota Fossil Reserve showing the principal excavation sites of the Alcoota Local Fauna.

Figure 2 Wakaleo alcootaensis, right dentary fragment (cast of UCMP 65621).

(A) Buccal view. (B) Occlusal view. (C) Lingual view. (D–F) Interpretive line drawings of (A–C) respectively. Note that specimen in (A–C) has been whitened with ammonium chloride. Abbreviations: ew, remnant enamel well; m1, first lower molar; m2, second lower molar; ta, talonid; tc, transverse crest; tr, trigonid. Scale bar = 20 mm.

During the 2013 field season a new pit was opened at Alcoota, on the same stratigraphic level as the other pits that quarry the Alcoota Local Fauna. This new pit, named ‘Shattered Dreams’, proved to be exceptionally densely packed with fragmented bones, interspersed with occasional complete, or near complete specimens. Not only was the volume of fossil bone extraordinarily high but so was the diversity, with virtually all known taxa from the Alcoota Local Fauna recovered from an area of less than two square meters. Included among these was a dentary belonging to W. alcootaensis (Figs. 3–5). This is the first substantial cranial specimen of this species found since the holotype was recovered 39 years previously. W. alcootaensis is now known from all of the main quarries of the Alcoota Local Fauna.

Figure 3 Wakaleo alcootaensis, incomplete right dentary, NTM P4325.

Photographs of the specimen after whitening with ammonium chloride. (A) Buccal view. (B) Lingual view. (C) Occlusal view. Scale bar = 50 mm.

Figure 4 Wakaleo alcootaensis, incomplete right dentary, NTM P4325.

Interpretive drawings of the photographs in Fig. 3. (A) Buccal view. (B) Lingual view. (C) Occlusal view. Abbreviations: bcp, base of the coronoid process; i1a, alveolus for first lower incisor; m1a, alveolus for first lower molar; m2a, alveolus for second lower molar; mfo, masseteric fossa; p3, third lower premolar; pmf, posterior mental foramen; s, symphyseal surface. Scale bar = 50 mm.

Figure 5 Wakaleo alcootaensis, right lower tooth row in occlusal view, NTM P4325.

(A) Photograph. (B) Interpretive drawing. Abbreviations: i1a, alveolus for first lower incisor; m1ara, alveolus for anterior root of first lower molar; m1pra, alveolus for posterior root of first lower molar; m2ara, alveolus for anterior root of second lower molar; m2pra, alveolus for posterior root of second lower molar; p3, third premolar. Scale bar = 20 mm.

In this paper all craniodental material of W. alcootaensis identified subsequent to the description of the holotype is described and illustrated. The diagnosis of W. alcootaensis is revised and new morphological character traits are identified that place the diagnosis on a firmer footing.

Geological Setting

The known fossils of W. alcootaensis all come from a dense bone bed in the lower part of the Waite Formation, cropping out on Alcoota Station, 110 km NE of Alice Springs in south central Northern Territory (Woodburne, 1967).The Waite Formation is a late Cenozoic sequence of fluviatile beds filling the Waite Basin, a small intermontane basin, surrounded by crystalline rocks of the Arunta Block (Woodburne, 1967). The Waite Formation consists of a basal series of overbank silts that were previously interpreted as lacustrine sediments (Woodburne, 1967) with interspersed and discontinuous carbonate-rich beds. The lower overbank beds are overlain by a coarser sequence of channel deposits, consisting of calcareous sandstones grading up into coarse red sandstones that contain a localised, silty, incised channel fill that is notable for containing the Ongeva Local Fauna (Megirian, Murray & Wells, 1997). The entire sequence is capped by a layer of silcrete. The bone bed that has produced the Alcoota Local Fauna and the W. alcootaensis fossils occurs in the lower overbank deposits, within a greyish-yellow silt unit that is interpreted as a crevasse-splay. The bone bed covers an area of approximately 25,000 m2, although its density and thickness varies considerably within that area (Megirian, 2000). The bulk of the known fossil material has been obtained from four pits: Paine Quarry, South Pit, Main Pit and Shattered Dreams (Fig. 1). The bone bed usually lies 90 cm below the present soil surface, underneath a reddish, weathered horizon (Murray & Megirian, 1992). It contains the unsorted but disarticulated and jumbled remains of many hundreds, if not thousands, of animals that appear to have perished in a mass death event, probably caused by severe drought (Murray & Vickers-Rich, 2004). Most of the in situ bones appear to be complete and show no signs of weathering prior to burial. Nevertheless the bones have undergone extensive fracturing due to the movements of the unconsolidated, clay-rich sediment that hosts them (Murray & Megirian, 1992).

Despite the apparently complete condition of most of the in situ bones, the known remains of W. alcootaensis are highly fragmented. In the case of the holotype the damage can be explained by the unfortunate circumstances of its discovery (Archer & Rich, 1982). In the case of the two dentary specimens it appears that both had weathered out of the primary bone bed and were broken up during their passage through the mobile cracking clays of the soil horizon that overlies the site.

It is thought that the fauna is late Miocene in age based on stage of evolution correlation using diprotodontid marsupials, (Stirton, Woodburne & Plane, 1967; Murray & Megirian, 1992), and its age lies between 5 and 12 ma (Megirian et al., 2010).

Methods

Terminology

Serial designation of the cheek dentition follows Flower (1867) and Luckett (1993). Standard nomenclature for mammalian tooth cusp anatomy is followed. Standard abbreviations for teeth are used: I, incisor; P, premolar; M, molar, with superscripts or subscripts representing upper or lower dentitions, respectively. Anterior and posterior are used as anatomical directions in the description of the dentition (instead of mesial and distal, respectively).

Measurements

Linear measurements were made with digital vernier callipers. Angular measurements were made with a protractor on a two-dimensional image taken normal to the plane of the angle being measured. The angle of the posterodorsal wall of the alveolus for I1 was measured by affixing a wooden splint flush against this wall with a small amount of petroleum jelly and measuring the angle of the protruding section.

Cladistic analysis

The broader intrafamilial relationships of Thylacoleonidae, particularly its basal branches are not examined here as the question has been comprehensively examined by Gillespie (2007) and will form the basis of a future publication. The present analysis is designed soley to test whether the new data provided here are enough to affect the position of W. alcootaensis, particularly in light of several derived conditions that are shared with the genus Thylacoleo. Only a single basal thylacoleonid, Priscileo roskellyae, is included to help polarise character states that vary between Wakaleo and Thylacoleo. Character state scores for this taxon were restricted to those that could be determined from available published descriptions and illustrations (Gillespie, 1997). The three named and currently accepted species of both Wakaleo (W. oldfieldi, W. vanderleueri and W. alcootaensis) and Thylacoleo (T. hilli, T. crassidentatus, and T. carnifex) form the rest of the ingroup (data sources in Table 1).

Table 1 Terminal taxa used in the cladistic analysis.

Outgroup and ingroup taxa with their sources for character data.

Taxon	Specimens examined	Literature used	
Pseudocheirus peregrinus	NTM U7839, U7840, U7841, U7843, U7846		
Nimiokoala greystanesi		Black & Archer, 1997	
Namilamadeta albivenator		Pledge, 2005	
Priscileo roskellyae		Gillespie, 1997	
Wakaleo oldfieldi	SAM P17925	Clemens & Plane, 1974; Gillespie et al., 2014	
Wakaleo vanderleueri	NTM P927-3, P8555-3, P8695-97, P87108-5, P87108-6	Megirian, 1986; Murray, Wells & Plane, 1987; Murray & Megirian, 1990; Gillespie et al., 2014	
Wakaleo alcootaensis	NTM P1, P4325, P4328, UCMP 65621 (c)	Archer & Rich, 1982	
Thylacoleo hilli		Pledge, 1977; Archer & Dawson, 1982	
Thylacoleo crassidentatus		Bartholomai, 1962; Archer & Dawson, 1982	
Thylacoleo carnifex		Owen, 1871; Owen, 1887; Archer & Dawson, 1982	

Three taxa were chosen to serve as serially distant outgroups: Namilamadeta albivenator, Nimiokoala greystanesi and Pseudocheirus peregrinus. Namilamadeta albivenator was chosen as a reasonably well-known, basal, non-thylacoleonid vombatimorphian (the sister group of Thylacoleonidae, Aplin & Archer, 1987). Nimiokoala greystenesi was selected as a basal member of Phascolarctimorphia, the sister group of Vombatimorphia. Pseudocheirus peregrinus is selected as a representative of Phalangerida, the sister group of Vombatiformes.

All ten terminal taxa were scored for 34 characters that were found to vary informatively within the restricted ingroup (Appendix 1). Characters were taken from Archer & Dawson (1982) and Gillespie (2007), with the addition of two novel characters.

Multistate characters that form obvious transformation series, such as the progressive enlargement of P3, were treated as ordered.

The resulting matrix was subjected to a maximum parsimony analysis in PAUP 4.0b (Swofford, 2002) using the following settings: heuristic search; random addition sequence with 500 replicates; and TBR branch-swapping algorithm. The strength of the internal nodes was tested with a decay analysis using the same settings.

Systematic Palaeontology

DIPROTODONTIA Owen, 1866	
VOMBATIFORMES Woodburne, 1984	
THYLACOLEONIDAE Gill, 1872	

Wakaleo alcootaensis Archer & Rich, 1982

Holotype. NTM P1, a fragment of a left maxilla, with P3 and fragments of M1 and M2. Found adjacent to Paine Quarry (Archer & Rich, 1982).

Referred material. NTM P4325, incomplete right dentary with broken P3 from Shattered Dreams that was originally assigned the unofficial field number BHP 12 (Figs. 3–5); NTM P4462, isolated right C1 from Main Pit (Fig. 6); NTM P4463; isolated right C1 from South Pit (Fig. 7); NTM P4328, isolated right M2 from Main Pit (Fig. 8); UCMP 65621, right dentary fragment with M1 and M2 from Paine Quarry (Fig. 2).

Figure 6 Wakaleo alcootaensis, photographs of isolated right upper canine, NTM P4462.

(A) Buccal view. (B) Anterior view. (C) Lingual view. (D) Posterior view. Scale bar = 10 mm.

Figure 7 Wakaleo alcootaensis, isolated right upper canine, NTM P4463.

(A) Photograph in buccal view. (B) Photograph in lingual view. (C) Photograph in occlusal view. (D) Interpretive drawing of (A). (E) Interpretive drawing of (B). (F) Interpretive drawing of (C). Abbreviations: ac, anterior carina; c, crown; pc, posterior carina; r, root. Specimen was photographed after being whitened with ammonium chloride. Scale bar = 20 mm.

Figure 8 Wakaleo alcootaensis, isolated left M2, NTM P4328.

(A) Photograph (stereopair) in occlusal view. (B) Interpretive drawing of occlusal view (note that this drawing was based on an earlier photograph and does not precisely match the photograph). (C) Photograph in buccal view. (D) Photograph in anterior view. (E) Interpretive drawing of (C). (F) Interpretive drawing of (D). Abbreviations: ant, anterior; ling, lingual; me, metacone; pa, paracone; pr, protocone; sb, vestigial stylar basin; tb, trigon basin. Scale bar = 10 mm.

Emended diagnosis

A species of Wakaleo distinguished from all others by: larger size (dental dimensions between 16 and 35% greater than the next largest species, W. vanderleueri, Table 2); anterior end of the masseteric fossa deeply recessed; long axis of I1 inclined at an angle greater than 50° to the horizontal ramus of the dentary; loss of P2; P3:M1 ratio of approximately 1.5; loss of M3.

Table 2 Measurements of dentaries and lower dentition of Wakaleo.

Measurements in mm.

	P3L	P3W	M1L	DH	P3–MF	P3–M2	M1–M2	
W. alcootaensis								
NTM P4325	19.6	(8.8)	∼13.1	30.4	53.8	42.0	24.0	
UCMP 65621	–	–	11.5	–	–	–	21.3	
W. vanderleueri								
NTM P85553-4	14.6	–	10.4	30.7	40.6	31.3	18.4	
NTM P8695-97	∼14.3	–	∼10.6	29.5	45.4	33.5	19.6	
NTM P87108-6	14.5	7.8	11.3	34.3	∼45.7	33.7	20.0	
NTM P87108-5	14.1	7.3	12.0	29.8	40.0	34.5	19.8	
NTM P9969-4	14.6	7.6		32.9	45.8	35.8	–	
NTM P2970-26	∼14.6	–	∼10.8	–	44.2	34.3	18.5	
MeanW. vanderleueri	14.5	7.6	11.0	31.4	43.6	33.9	19.3	
W. oldfieldi								
SAM P17925	12.5	7.8	10.5	26.8	39.2	31.8	–	
Notes.

L anteroposterior length of crown

W maximum buccolingual width of the crown

DH dentary height measured at the level of the posterior margin of M2

P3 −−MF distance from the posterior margin of the P3 to the anterior rim of the masseteric fossa

P3 −−M2 length of the tooth row from the anterior margin of P3 to the posterior margin of M2

M1 −−M2 combined length of M1 and M2

Description

Dentary. NTM P4325 (Figs. 3–5) is similar in size and shape to the larger dentaries of W. vanderleueri (e.g., NTM P87108-6). The horizontal ramus deepens anteriorly to reach a maximum depth under the midlength of the P3. The ventral border forms a straight line for its entire length. It is moderately thick buccolingually, with a midlength width of 14.1 mm and a mild dorsoventral convexity on the buccal surface. Species of Wakaleo bear a large anterior mental foramen on the buccal surface of the dentary adjacent to the incisor, and one or two smaller accessory mental foramina posterior to it. The external opening of the anterior mental foramen in NTM P4325 is missing due to damage but a single small posterior mental foramen is present, ventral to the posterior root of P3. The lingual surface of the dentary bears a shallow, narrowly triangular, digastric fossa impressed upon the posterior half of the horizontal ramus. Only the posterior end of the symphyseal surface is present, it extends posteriorly to the level of the middle of P3. In occlusal view the longitudinal axis of the horizontal ramus is inclined at an angle of 20° to the symphyseal plane.

Although the anterior end of the dentary is missing, the posterodorsal wall of the alveolus for I1 is preserved. It indicates that the incisor projected at an angle of 54° from the longitudinal axis of the dentary. A very short diastema separates this alveolus from the alveolus for P3. There is no alveolus for any rudimentary teeth between I1 and P3. The P3 dominates the dentary, occupying 45% of the total length of the cheek tooth row, or 35% of the distance from the anterior margin of the masseteric fossa to the anterior end of the P3. Unfortunately the crown is largely broken away, preventing description of this tooth beyond its size. In occlusal view the anterior end of the P3 can be seen to be angled lingually so that the distance between the symphyseal plane and the anterior end of P3 is less than the distance from the symphyseal plane to the posterior end of P3 (8.2 mm vs. 12.8 mm). Four alveolar sockets follow the P3 in a linear row without any diastemata. Since the lower molars of Wakaleo are known to be double rooted, it is clear that there were only two molars present behind P3. Thus, M3 was absent in this species. Although the buccal alveolar margin of NTM P4325 is partially eroded, it is clear that the lingual margin was higher and the alveoli were canted to face slightly buccally. This is reflected in the molar crowns of UCMP 65621 which are angled so that the occlusal surfaces face buccodorsally. In lingual view the alveolar margin slopes ventrally from posterior edge of P3 to the second root socket of M1, after which it levels out and becomes roughly horizontal. The only known lower molars of W. alcootaensis are the two present in UCMP 65621 (Fig. 2). These are heavily worn and present few details. M1 resembles the M1 of other Wakaleo species in having a subrectangular occlusal outline, with a weak constriction separating the trigonid from the talonid. The lingual half of the trigonid is missing, including the large anterior cusp present in other Wakaleo species. However the buccal side of the trigonid bears a transverse crest that rises lingually to meet this cusp as in the M1 of other species of Wakaleo. The anteroposterior length of the talonid is approximately equal to that of the trigonid. It has a squared-off posterior margin in occlusal view like other Wakaleo species. Although heavily worn, there is a small well of enamel remaining in the centre of the talonid basin (Woodburne, 1967). It is smooth but may be too small and worn to accurately determine if crenulations were present or absent. The trigonid and talonid are supported by a single root each. The exposed trigonid root is anteroposteriorly compressed and buccolingually expanded in cross section. The partially obscured talonid root has an anteroposteriorly thicker cross-section than the trigonid root.

M2 is smaller with a more rounded occlusal outline. The trigonid is less strongly raised than in M1. It bears a low transverse ridge. Unlike M1 the talonid of M2 has a rounded posterior margin. The talonid is narrower than the trigonid although its length is approximately the same as that of the trigonid. As in M1, the talonid basin bears a small remnant well of enamel with a smooth surface.

Posterior to the molar tooth row the dentary rises quickly into the ascending ramus, with almost no postalveolar shelf. The buccal surface of base of the ascending process is deeply excavated by a sharply defined masseteric fossa. The anterior end of the masseteric fossa is recessed for several millimetres under the anterior rim forming a blind pocket. This recess is deeper than in the holotype of W. oldfieldi. The anterior margin of the ascending process is supported by a spar-like rib that is transversely broader than it is anteroposteriorly deep. This rib forms the anterodorsal margin of the masseteric fossa.

Upper dentition. Two isolated upper canines are known (Figs. 6 and 7). Each has a single root and a unicuspid crown. The crown is angled lingually relative to the root so that the apex lies level with the lingual side of the crown in occlusal view (NTM P4463, Fig. 7) or overhangs it (NTM P4462, Fig. 6). The long axis of the crown in buccal view is angled posteriorly relative to the root, it is lingually inclined in anterior and posterior views. The root is complete in NTM P4463 (Fig. 7). It is roughly banana-shaped and is approximately four times longer than the crown. It tapers to point at its base and expands to a maximum thickness of 10 mm, at 17 mm from the base. It is gently constricted below the base of the crown, forming a neck that is slightly narrower than the base of the crown in buccal and lingual view.

The crown is spade-shaped in buccal view with a bluntly-rounded apex. The height of the crown is approximately equal its anteroposterior basal length. The anterior and posterior margins bear carinae that extend from the base to the apex, meeting at its tip and dividing the crown into distinct buccal and lingual faces. The buccal face is more distinctly convex in transverse section than the flattened lingual face. The measurements of these canine crowns (Table 4) match those reported for W. vanderleueri (Gillespie, 2007), and are smaller than the canine alveolar dimensions of CPC 26604. Given that all other dental specimens of W. alcootaensis show that it had dimensions in excess of those of W. vanderleueri, it would appear that the canines of W. alcootaensis were reduced relative to its other teeth in comparison to the former species.

Table 3 Measurements of the second upper molar of Wakaleo alcootaensis and W. vanderleueri.

Measurements in mm.

	M2L	M2L (roots)	M2W	
W. alcootaensis				
NTM P1	–	7.2	9.0	
NTM P4328	9.4	8.0	9.1	
W. vanderleueri				
NTM P87103-9	7.5	6.7	8.4	
CPC 26604	7.0	–	9.5	
Notes.

L maximum anteroposterior length of the crown

L(roots) minimum anteroposterior length, measured at the constriction below the crown

W maximum buccolingual width of the crown

Table 4 Measurements of the upper canine of Wakaleo alcootaensis.

Measurements in mm.

	L	W	H	RootL	
NTM P4462	7.0	4.9	7.1	–	
NTM P4463	7.9	5.8	7.4	25.9	
Notes.

L maximum anteroposterior length of the crown

W buccolingual width of the crown at its base

H height of the crown from base to apex

RootL length of the root from its tip to the base of the crown

The second upper molar of the holotype is only represented by broken roots in the alveolus but an isolated left M2 is now known (Fig. 8). It is slightly larger than the M2 of the holotype. Note that the anteroposterior length of the buccal side appears to be significantly greater than the measurement reported in Archer & Rich (1982), but this is because the crown is wider than the roots. The crown of M2 is missing in the holotype and the length measurement was obtained from the distance between the anterior and posterior roots . When the same measurement is taken on NTM P4328, the difference in length between the two specimens is less than 12% (Table 3). The crown is distinctly trigonal and tritubercular. The occlusal outline of the tooth is nearly equilateral with its buccolingual width similar to its anteroposterior length. This differs from the M2 of W. vanderleueri in which the width is distinctly greater than the length. Note that because of the equilateral nature of NTM P4328, no side is significantly longer than any other and this difference cannot be explained by a misinterpretation in the orientation of the tooth. A very weak flexus causes a slight emargination on the anterolingual side of the crown in occlusal view, possibly where the posterior margin of M1 impinged upon M2 as it does in W. vanderleueri (Murray, Wells & Plane, 1987, Fig. 8). A small depression, containing some grains of adherent matrix lies close to the anterobuccal margin, just anterior and slightly buccal to the metacone, and is interpreted as a vestigial stylar basin. A low cusp is developed at each corner of the crown. These three cusps are the paracone, metacone and protocone. There is no trace of a metaconule. Of these three cusps the paracone is the tallest, represented by a low peaked ridge. The peak is inset from the buccal margin, and the crown is expanded laterally between the roots and the peak of the paracone. However abrasion of the enamel along the buccal margin means that it is not possible to see if the lateral bulge above the paracone is as well developed as it is in W. vanderleueri (Gillespie, 2007). The height of the cusp is not as great relative to the other cusps as it is in W. vanderleueri where the paracone forms a tall peak, even in worn specimens (e.g., NTM P87103-9). Indeed the entire buccal margin is of a similar depth to the lingual margin, unlike the condition in W. oldfieldi and W. vanderleueri where the buccal margin is distinctly deeper than the lingual margin (Gillespie, 2007). Both the protocone and the metacone of NTM P4328 have been worn virtually flat. Low rounded crests connect each cusp and define a triangular, smooth trigon basin that dominates the occlusal surface of the tooth. No crenulations are present in this basin. The deepest point of the basin is slightly off-centre and located closer to the posterolingual rim than the other two sides. A poorly defined shallow trough that extends along the inside of the anterolingual rim of the trigon basin is interpreted as a feature caused by wear. The crown is supported by three subequal roots developed at each of the corners of the trigon. The roots are directed posteriorly and lingually relative to the plane of the crown as in Wakaleo vanderleuri (NTM P87103-9). The root orientation and presence of a vestigial stylar shelf indicate that this equilateral and almost triradially symmetric tooth has been correctly oriented.

Discussion

Referral of the new material to Wakaleo alcootaensis

The dentaries described here can be referred to Thylacoleonidae on the basis of the enlarged P3 (NTM P4325), strong reduction in the size of the posterior molars (UCMP 65621 and NTM P4325) and posterior narrowing of the lower molars (UCMP 65621). Within Thylacoleonidae the specimens can be referred to Wakaleo by the loss of the anterior premolars (NTM P4325). Both of these specimens exceed the size of W. vanderleueri and W. oldfieldi and have apparently lost M3, excluding them from either species. They can be referred to W. alcootaensis on the basis of matching large size and co-occurrence with the holotype.

The simple, flattened tritubercular and triangular upper molar (NTM P4328), lacking a metaconule and with only a vestigial trace of a stylar shelf, strongly resembles the more posterior upper molars of other species of Wakaleo as opposed to the more rectangular molars of Priscileo and Thylacoleo. As in the dentaries, the matching size of this specimen (Table 3) and its co-occurrence with the holotype of W. alcootaensis, indicate that it can be referred to this species.

As yet, no descriptions of the canines of other Wakaleo species have been published. Nonetheless canines are known for an unnamed, primitive species of Wakaleo and W. vanderleueri both of which have been described in an unpublished PhD thesis (Gillespie, 2007). Those specimens agree with the canines described here in all salient features, including the rounded apex, gentle recurvature and carinate anterior and posterior edges dividing a more convex buccal face from a flatter lingual face. No other mammal known from the Alcoota Local Fauna has a tooth with this combination of features. Thus these isolated specimens can be referred to Wakaleo. They can be referred to the species W. alcootaensis on the basis of co-occurrence.

It remains a far simpler explanation of the data that all of the large-sized Wakaleo fossils at Alcoota belong to a single species rather than to posit multiple large-bodied thylacoleonid taxa for which there is no evidence. In other assemblages where there are two co-occurring thylacoleonid species there is always a large size difference between them. For example the small house cat sized Priscileo roskellyae co-occurs with the leopard sized Wakaleo oldfieldi in the middle Miocene Cleft of Ages Local Fauna of Riversleigh (Archer et al., 2006; Gillespie, 2007) and Thylacoleo hilli has dental dimensions half those of T. crassidentatus with which it co-occurs in the Pliocene Bow Local Fauna (Archer & Dawson, 1982). Furthermore, most of the specimens discussed here show characteristics that are diagnostic of the genus Wakaleo. Since no more than one Wakaleo species is ever present in any one local fauna (Gillespie, 2007), this observation adds further support to the hypothesis that the entire sample belongs to a single species.

Diagnostic characters of W. alcootaensis

Wakaleo alcootaensis was originally diagnosed as distinct from W. oldfieldi and W. vanderleueri on the basis of size, with the P3 reaching approximately twice the length of W. oldfieldi (Archer & Rich, 1982) and maxillary dimensions that are about 30% larger than those of W. vanderleueri (Murray & Megirian, 1990). Murray & Megirian (1990) suggested that apart from its larger size, W. alcootaensis lacks any significant morphological differences from the smaller, older species of Wakaleo. With the addition of further specimens, including lower jaws, the diagnosis of W. alcootaensis can be expanded. Several characters can now be seen to differentiate the admittedly meagre W. alcootaensis material from the other two named species of the genus (Fig. 9). These are listed briefly above and, given that some interpretation is required, discussed in more detail here:

Figure 9 Comparison of the dentaries of Wakaleo species.

(A) W. alcootaensis. (B) W. vanderleueri. (C) W. oldfieldi. Differences between the species illustrated are: the depth of the anterior recess of the masseteric fossa; the presence or absence of P2 and M3, the ratio of the length of M1 to P3, and the angle of the base of I1 relative to the horizontal axis of the dentary. (A) is reconstructed from NTM P4325 and UCMP 65621. (B) reconstructed from NTM P87108-5 and NTM P87108-6. (C) redrawn from Clemens & Plane (1974, Fig. 1A, reversed for comparison). Scale bar = 40 mm.

Larger size. As can be seen from the measurements in Table 2 and the discussion on variation below, the size of all known specimens of W. alcootaensis exceeds the known range of W. vanderleueri and W. oldfieldi in almost all dimensions. The only measurement for which NTM P4325 falls within the range of W. vanderleueri is dentary height (measured as the dorso-ventral height of the dentary at the posterior end of M2), indicating that the species, or at least this individual, was somewhat slender jawed when compared to the most robust individuals of W. vanderleueri (e.g., NTM P87108-6).

Deeply recessed masseteric fossa. The anterior margin of the masseteric fossa of W. vanderleueri varies from a gentle change in slope of the buccal surface of the dentary resulting in a bevelled margin (e.g., NTM P87108-6) to a sharply impressed fossa with the anterior margin forming low walls perpendicular to the buccal surface. In NTM P4325 the nature of the fossa resembles the latter condition but the anterior end of the fossa is recessed under its rim, forming a blind pocket (Fig. 3A). The same condition was described in the holotype of W. oldfieldi (Clemens & Plane, 1974), although inspection of this specimen by the author reveals that the recess is barely developed and much shallower than it is in NTM P4325.

Steeply-angled I1. The basal section of the lower incisor of the holotype of W. oldfieldi projects anterodorsally at an angle of 27° from the long axis of the horizontal ramus of the dentary before the apical region of the tooth curves dorsally (Fig. 10). The only I1 of W. vanderleueri in place in a jaw (NTM P87108-5) is similarly procumbent with an angle of 31°. In other specimens where the I1 is missing (NTM P9273-3, P85553-4, P8695-97, P87108-6) the angle of the posterodorsal wall of the alveolus can be measured. This angle ranges from 30 to 38° in these specimens with a mean of 34.8° , indicating the posterodorsal wall of the alveolus is an acceptable proxy for the angle of procumbency of I1. In contrast the posterodorsal wall of the alveolus for I1 of W. alcootaensis (NTM P4325) is far more steeply-angled at close to 54° from the long axis of the horizontal ramus (Fig. 10).

Figure 10 Inclination of the first lower incisor in Wakaleo dentaries.

(A) Wakaleo oldfieldi, (B–F), Wakaleo vanderleueri. (G) Wakaleo alcootaensis. (E) and (G) reversed for comparison. Specimens missing the first lower incisor have a splint affixed to the posterodorsal wall of the alveolus. Note the steep inclination in W. alcootaensis. Drawings from photographic images, not to scale.

Loss of P2. The holotype of W. oldfieldi and all specimens of W. vanderleueri that preserve the bone between I1 and P3 retain a rudimentary single cusped tooth, or an alveolus for such a tooth (Clemens & Plane, 1974; Megirian, 1986; NTM P927-3, NTM P8695-97, NTM P-87108-6). This tooth is usually identified as P2 (e.g., Megirian, 1986; Murray, Wells & Plane, 1987) although other identifications, such as P1 or a canine, are possible. Although the presence of an upper anterior premolar is variable within W. vanderleueri (Murray & Megirian, 1990) it would appear that a lower tooth in this position is invariably present. The Alcoota dentary lacks any alveolus between I1 and P3 (Figs. 3A and 3C) indicating the complete loss of all lower cheek teeth anterior to P3.

Larger P3 relative to M1. The P3:M1 length ratio for W. oldfieldi is 1.19 in the holotype and 1.16 in a specimen from Riversleigh (Gillespie et al., 2014). This ratio ranges from 1.18 to 1.40 in the Bullock Creek sample of W. vanderleueri (SAM P17925, NTM P2970-26, NTM P87108-5, NTM P87108-6, NTM P85553-4). The precise ratio in NTM P4325 cannot be obtained because M1 is missing and its length has to be taken from that of its alveolus. If this is done, a ratio of 1.50 is obtained. Thus, even allowing for estimation errors, it is clear that W. alcootaensis has a distinctly larger P3 to M1 ratio than W. oldfieldi and one that lies outside the range of variation seen in W. vanderleueri. Although the length of M1 has to be estimated in NTM P4325 it is possible to compare the size of P3 with a measureable proxy for total jaw size. If P3 is compared to the distance from the anterior margin of the alveolus for P3 to the anteriormost point of the masseteric fossa, similar results to the comparison of P3 and M1 are obtained. P3 is 30.7% of the jaw size proxy in the holotype of W. oldfieldi, while it ranges from 31.5% to 36.0% in W. vanderleueri and is 36.4% in NTM P4325. Thus W. alcootaensis has a distinctly enlarged P3 in comparison with W. oldfieldi and a slightly enlarged P3 in comparison with W. vanderleueri.

Loss of M3. The lower jaws of W. oldfieldi and W. vanderleueri bear three double-rooted molars behind the enlarged P3 (Clemens & Plane, 1974; Megirian, 1986; Gillespie et al., 2014). The new Alcoota dentary (NTM P4325) bears just four sockets (Fig. 5), indicating only two double-rooted molars. An alternative interpretation was suggested during the review of this paper. In this interpretation the tall peak of alveolar bone observed at the anterior end of NTM P4325 is taken to mark the boundary between the last premolar and the first molar. This allows a linear row of five sockets for the roots of the molar teeth which would presumably be interpreted as receiving two double rooted molars and a posterior single rooted molar. However such an interpretation can be dismissed because the broken roots in the first two alveolar sockets form a contiguous broken surface over the peak of alveolar bone (Figs. 4C, 5 and 11), indicating conclusively that they are the anterior and posterior roots of the same large premolar. Furthermore the tall peak of alveolar bone matches precisely the peak that occurs between the anterior and posterior roots of P3 in W. vanderleueri (Fig. 11). That the four remaining alveolar sockets equate to two double rooted molar teeth is supported by the presence of two roots in all known Wakaleo lower molars, including the reduced M3 of W. vanderleueri (Clemens & Plane, 1974; Gillespie et al., 2014). This interpretation is further supported by the dentary fragment UCMP 65621, which preserves its last two molars. These molars would appear to be homologous with M1 and M2 of W. vanderleueri, indicating that M3 was absent in this specimen as well. The posterior molar of UCMP 65621 is identified as M2 rather than M3 because it is much larger than the reduced M3 of other Wakaleo species both in terms of absolute size and relative size compared to the preceding molar. It also retains distinct trigonid and talonid moieties unlike the M3 of W. oldfieldi or W. vanderleueri. In W. oldfieldi the talonid basin occupies most of the occlusal surface of the tooth, with the trigonid reduced to a raised anterior edge or absent altogether, while in W. vanderleueri reduction of M3 has proceeded to the point that it is a simple basinless nubbin.

Variation within Wakaleo alcootaensis

As there is very little overlap between the new specimens and the holotype any discussion of variation within W. alcootaensis is restricted to size variation. The holotype has dental measurements that are about one third larger than those of W. vanderleueri. Similarly the new dentary has dental measurements that range from 16 to 35% greater than the mean value for W. vanderleueri (Table 2). However the dentary fragment UCMP 65621 is not so large, with the combined length of M1 and M2 only exceeding the mean value for W. vanderleueri by just over 10% (Table 2). In contrast, the isolated M2 is slightly larger than that of the holotype of W. alcootaensis, although the difference is less than 15% of the linear measurements. These observations indicate that, like W. vanderleueri, W. alcootaensis displayed a modest range of size variation.

Evolution within Thylacoleonidae

Members of the genus Wakaleo, including W. alcootaensis, display several derived features not seen in species of Thylacoleo such as loss of the first premolar in the upper and lower jaws, presence of an anterolingual cuspule on the third upper premolar and triangular upper molars (Gillespie, 2007). These suggest that Wakaleo forms a clade to the exclusion of Thylacoleo as suggested by Clemens & Plane (1974). Nonetheless W. alcootaensis displays several derived states not present in earlier Wakaleo species but are present in Thylacoleo. These include: larger size; steeply-angled lower incisors; increased size of P3 relative to other teeth; and reduction in the number of molar teeth. To test whether or not these character states are sufficient to remove W. alcootaensis from Wakaleo, or to nest Thylacoleo within Wakaleo as the sister taxon of W. alcootaensis, a cladistic analysis was performed. The search produced three most parsimonious trees with a length of 63 steps. The strict consensus of these trees upholds Wakalaeo as a clade including W. alcootaensis (Fig. 12).

Figure 11 Comparison of Wakaleo dentaries showing P3.

(A) Anterior end of left dentary of W. vanderleueri (NTM P927-3) in buccal view. (B) Anterior end of right dentary of W. alcootaensis (NTM P4325) in buccal view (reversed for comparison). Long arrows point to tall process of alveolar bone dividing the anterior and posterior roots of P3. Short arrow in (B) represents continuation of broken dentine over the top of the process demonstrating that the first two sockets belong to the same tooth. Scale bars = 20 mm.

Figure 12 Strict consensus tree of two most-parsimonious-trees obtained from cladistic analysis of thylacoleonid interrelationships.

Source trees have a length of 63 steps. Numbers with node represent decay index values.

Of the features shared between W. alcootaensis and Thylacoleo that were included in the analysis (that is all except absolute size) all were optimised as convergences between W. alcootaensis and a subset of Thylacoleo (T. crassidentatus + T. carnifex), or T. carnifex alone, at least in delayed transformation optimisation. None of them were found to be synapomorphies linking W. alcootaensis to Thylacoleo. The enlargement of P3 and the loss of M3 were interpreted as synapomorphies of Thylacoleonidae that were reversed in W. oldfieldi and W. vanderleueri when acctran optimisation was in place. This optimisation, although equally parsimonious within the narrow parameters of the present analysis, is incongruent with stratigraphy and is almost certainly an artefact of the high amounts of missing data for basal thylacoleonids. As new data for Priscileo roskellyae and other basal thylacoleonids become available it is likely that the ambiguity will be resolved in favour of the deltran optimisation. Thus the interpretation that the similarities between W. alcootaensis and Thylacoleo are convergent is supported by the analysis, although the strength of this support is lessened by missing data. This indicates that there has probably been a certain amount of iterative evolution in Thylacoleonidae with some character traits evolving in the late Miocene of the Wakaleo clade and again, independently, in the Plio-Pleistocene Thylacoleo clade. What selective force may be driving this convergence is unknown, although the increased size of both W. alcootaensis and later Thylacoleo, relative to other thylacoleonids hints that it may be a specialisation towards hypercarnivory and increasing prey size.

The new anatomical information and the phylogenetic analysis also allow us to revisit the position of W. alcootaensis within Wakaleo. Previous hypotheses had suggested that Wakaleo consisted of a single anagenetic lineage passing from W. oldfieldi to W. vanderleueri and finally W. alcootaensis. This hypothesis is supported by the stratigraphic succession of these taxa and the apparent morphoclinal trends that they exhibit. ‘Apparent’ is an appropriate qualifier because the incompleteness of both W. oldfieldi and W. alcootaensis meant that no single anatomical structure could be traced through all three species. With the addition of upper jaw material for W. oldfieldi (Gillespie et al., 2014) and lower jaw material for W. alcootaensis (this paper) these morphoclinal trends can be re-examined. The following character trends are found to be congruent with an anagenetic lineage: increasing absolute size; increasing P3 to M1 ratio; and progressive reduction and eventual loss of M3. We might also add an increasingly steeply inclined I1 if it can be shown that like the holotype other W. oldfieldi individuals have a highly procumbent I1 set at a lower angle to those of W. vanderleueri. However other characters are incongruent with this morphoclinal trend. Incongruent characters include the buccal height of M2 relative to its lingual height and the excavation of the anterior margin of the masseteric fossa. W. oldfieldi and W. vanderleueri show increasing height of the buccal side of M2 relative to the lingual side so there is a steep buccolingual gradient across the tooth. In contrast the buccal side of the M2 of W. alcootaensis is barely any taller than the lingual side (NTM P4328). The masseteric fossa is recessed under its anterior rim in W. oldfieldi and W. alcootaensis whereas there is no recess in W. vanderleueri. These characters may be simply represent small-scale reversals within an anagentic lineage or may be indicative of a more complex branching arrangement within Wakaleo.

Although simple cladistics analysis is incapable of testing for anagenesis, with each operational taxonomic unit treated as a terminal branch, we can expect that anagenetic lineages appear as a pectinate arrangement with the constituent taxa appearing in sequence. This does not occur within the Wakaleo clade of the present analysis, casting some doubt upon the hypothesis of an anagenetic lineage in this genus. According to the analysis W. alcootaensis branched off prior to the split between W. oldfieldi and W. vanderleueri. This implies a ghost lineage for W. alcootaensis extending back to the early Miocene, for which we have no physical evidence. However an examination of the synapomorphies supporting the W. oldfieldi + W. vanderleueri clade shows that they are mostly plesiomorphic characters that have been optimised as reversals in this analysis. This may well be an artefact of the poor representation of basal thylacoleonids in the analysis and will likely change with the addition of new, currently unpublished, basal thylacoleonid material (Gillespie, 2007).

In summary, W. alcootensis would appear to be correctly placed in Wakaleo, which is supported as monophyletic but an evaluation of evolution within the genus is dependent upon the addition of new data, much of which should be forthcoming.

Supplemental Information

Supplemental Information 1 Character-taxon matrix in nexus format

Click here for additional data file.

The new dentary which form the basis of this paper was discovered in a new pit (‘Shattered Dreams’) which was opened thanks to the generous loan of a backhoe and licensed operator from Central Desert Regional Council, Northern Territory. I am deeply indebted to Glenn Marshall for making this loan possible. I also wish to thank Ben McHenry of the South Australian Museum for allowing me access to thylacoleonid specimens in his care. The photographs in Figs. 2–5 were taken by Steven Jackson. Jay Nair assisted in the location of some literature. The manuscript was greatly improved by thorough reviews from Kenny Travouillon, Maria Amelia Chemisquy, Graciela Piñeiro and an anonymous reviewer.

Institutional Abbreviations

CPC Commonwealth Palaeontological Collection, Geoscience Australia, Canberra

NTM Museum and Art Gallery of the Northern Territory, Darwin and Alice Springs

SAM South Australian Museum, Adelaide

UCMP Museum of Paleontology, University of California, Berkeley

Appendix 1. Character List

1. Number of alveoli between P3 and I1: no alveoli (0); one alveolus (1); two alveoli (2) (modified from characters 1 and 2 of Gillespie, 2007). Character is treated as ordered. Given the difficulty in determining the homology of the reduced anterior teeth between the first incisor and the third premolar in diprotodontians, this character is simplified here and simply codes the variable number of alveoli present between these teeth.

2. Ratio of the length of P3to the length of M1: less than 1.0 (0); between 1 and 1.1 (1); between 1.1 and 1.5 (2); 1.5 or greater (3) (modified from Clemens & Plane, 1974). The character is treated as ordered.

3. Development of a posterolingual crest on P3: weakly developed to absent (0); well-developed (1) (modified from character 7 in Gillespie, 2007). The character here is simplified by subsuming the state of ‘weakly developed’ into the plesiomorphic state.

4. Presence or absence of a weak anterobuccal crest on P3: absent (0); present (1) (modified from character 9 in Gillespie, 2007). The character here is treated as a simple presence or absence character, rather than distinguishing between weakly and moderately developed derived states.

5. Molar cusp morphology: selenodont (0); bunolophodont (1); bunodont (2) (from Gillespie, 2007).

6. Height of the trigonid relative to the talonid in M1: trigonid subequal or lower than the talonid height (0); trigonid distinctly taller than the talonid (1); trigonid more than twice the height of the talonid (2) (modified from character 12 in Gillespie, 2007). Character is ordered.

7. Width of talonid basin of M1: width nearly equal to width of the crown (0); width less than 70% of the width of the crown (1); width reduced to less than 30% of the width of the crown, or near absence (2) (modified from character 13 in Gillespie, 2007). Character is ordered.

8. Width of the talonid moiety relative to the trigonid moiety in M1: talonid wider than the trigonid (0); talonid between 70% and 100% the width of the trigonid (1); talonid less then 70% of the width of the trigonid (2) (character modified from character 14 in Gillespie, 2007). Character is ordered.

9. Height of the trigonid relative to the talonid in M2: trigonid slightly taller than the talonid height (0); trigonid greater than 1.3 times taller than the talonid (1) (modified from character 16 in Gillespie, 2007).

10. Presence or absence of the talonid in M2: present (0); absent (1) (modified from character 17 in Gillespie, 2007).

11. Presence or absence of M3: present (0); absent (1) (from Archer & Dawson, 1982).

12. Presence or absence of M4: present (0); absent (1) (from Archer & Dawson, 1982).

13. Ratio of the length of P3 to M1: less than 1.0 (0); between 1 and 1.5 (1); between 1.5 and 1.9 (2); greater than 1.9 (3) (modified from character 23 in Gillespie, 2007). Character is ordered.

14. Length of the longitudinal blade of P3 relative to the total length of the tooth: less than 50% (0); between 50 and 70% (1); greater than 70% (2) (from Gillespie, 2007). Character is ordered.

15. Orientation of posterior longitudinal blade of P3: steeply inclined (0); gently bowed and horizontal (1); absent (2) (from Gillespie, 2007).

16. Presence or absence of a mid crown constriction of P3: absent (0); present (1) (from Gillespie, 2007).

17. Presence or absence of a posterobuccal crest on P3: absent (0); present (1) (from Gillespie, 2007).

18. Shape of the prominence below the anterior cusp of P3 on its lingual side: elongate crest joining anterior cusp (0); cuspule sepparated from anterior cusp (1) (from Gillespie, 2007).

19. Posterior width of P3 relative to anterior width: greater than anterior width (0); less than anterior width (1) (from Gillespie, 2007).

20. Occlusal outline of M1: square to rectangular (0); triangular (1) (from Gillespie, 2007).

21. Development of the metaconule of M1: present and well-developed (0); barely developed or absent altogether (1) (from Gillespie, 2007).

22. Presence or absence of P1: present (0); absent (1) (from Gillespie, 2007).

23. Presence or absence of P2: present (0); absent (1) (from Gillespie, 2007).

24. Shape of longitudinal crest of P3 in occlusal view: straight (0); longitudinally bowed (1) (from Gillespie, 2007).

25. Shape of posterobuccal margin of M1 and relationship to M2: not elongated and not overlapping M2 (0); posteriorly elongated and overlapping M2 in lateral view (1) (modified from character 33 in Gillespie, 2007).

26. Anteroposterior depth gradient of the buccal side of the crown of M1: no gradient, anterior and posterior ends of the crown of equal depth (0), weak gradient with anterior end slightly taller than the posterior end (1), strong gradient with anterior end much taller than the posterior end (2) (from Gillespie, 2007). Character is ordered.

27. Occlusal outline of M2: rectangular (0); subtriangular (1); fully triangular (2) (from Gillespie, 2007). Character is ordered.

28. Presence or absence of a metaconule on M2: present (0); absent (1) (from Gillespie, 2007).

29. Presence or absence of a lateral bulge of the buccal margin of the crown, adjacent to the paracone of M2: absent (0); present (1) (from Gillespie, 2007).

30. Presence or absence of M3: present (0); absent (1) (from Archer & Dawson, 1982).

31. Angle of the long axis of I1 to the long axis of the horizontal ramus of the dentary: less than 40° (0); greater than 40° (1). Character is new.

32. Development of the masseteric fossa: fossa is not recessed under the anterior margin (0); fossa is recessed under the anterior margin (1). Character is new.

33. Presence or absence of palatal ridges: absent (0); present (1) (from Gillespie, 2007).

34. Depth of buccal margin of M2 in comparison to the lingual margin: buccal and lingual margins of similar height (0); buccal margin raised relative to the lingual margin (1) (from Gillespie, 2007).

Appendix 2. Tree Description

The second of two most-parsimonious-trees (where Thylacoleo hilli is resolved as the sister taxon of T. crassidentatus + T. carnifex) is described, however only ingroup clades that are present in the strict consensus tree are described. State changes are given in brackets. An asterisk denotes a state change that occurs once, without homoplasy.

Clade 1. Thylacoleonidae

Unambiguous synapomorphies. Character 5 (1 to 2, or 0 to 2)*, bunodont molar cusps. Acctran optimisation supports the state change from bunolophodont to bunodont (1 to 2), whereas deltran optimisation interprets the state change as selenodont to bunodont (0 to 2). In either case the change to bunodonty is unambiguously tied to this clade and is an unambiguous synapomorphy of Thylacoleonidae. Character 15 (0 to 1)*: posterior longitudinal blade of P3 is nearly horizontal and bowed. Character 24 (0 to 1)*: main sectorial blade of P3 is longitudinally bowed. Character 26 (0 to 1)*: an anteroposterior gradient on the buccal side of M1 with a taller anterior end. Character 29 (0 to 1)*: presence of a lateral bulge of the buccal margin of M2 adjacent to the paracone.

Ambiguous synapomorphies under acctran optimisation. Character 2 (1 to 3): a P3to M1 ratio of 1.5 or more. Unknown in Priscileo roskellyae and reversed in Wakaleo oldfieldi + Wakaleo vanderleueri. Deltran optimisation interprets this chacter state as a convergence between Wakaleo alcootaensis and Thylacoleo crassidentatus + Thylacoleo carnifex. With the inclusion of new data from basal thylacoleonids this ambiguity will almost certainly resolve in favour of the deltran optimisation. Character 3 (0 to 1): presence of a well-developed posterolingual crest on P3 (reversed in Thylacoleo crassidentatus + Thylacoleo carnifex). Deltran optimisation interprets this character as a convergence between Wakaleo oldfieldi + Wakaleo vanderleueri and Thylacoleo hilli. Character 6 (0 to 1)*: trigonids of lower molars taller than the talonids. Ambiguous due to missing published information on the lower molars of Priscileo roskellyae. Deltran optimisation finds this character state change on the branch supporting Wakaleo + Thylacoleo. Character 7 (0 to 1)*: talonid basins of lower molars distinctly narrower than the crown. Ambiguous due to missing published information on the lower molars of Priscileo roskellyae. Deltran optimisation finds this character state change on the branch supporting Wakaleo + Thylacoleo. Character 8 (0 to 1)*: talonid moiety of M1 narrower than the trigonid moiety. Ambiguous due to missing published information on the lower molars of Priscileo roskellyae. Deltran optimisation finds this character state change on the branch supporting Wakaleo + Thylacoleo. Character 9 (0 to 1)*: trigonid of M2 much higher than its talonid. Ambiguous due to missing published information on the lower molars of Priscileo roskellyae. Deltran optimisation finds this character state change on the branch supporting Wakaleo (the state cannot be determined in Thylacoleo due to loss of the talonid in M2). Character 11 (0 to 1): loss of M3 (reversed in Wakaleo oldfieldi + Wakaleo vanderleueri). Deltran optimisation interprets this chacter state as a convergence between Wakaleo alcootaensis and Thylacoleo crassidentatus + Thylacoleo carnifex. With the inclusion of new data from basal thylacoleonids this ambiguity will almost certainly resolve in favour of the deltran optimisation. Character 12 (0 to 1)*: loss of M4. Ambiguous due to missing data for Priscileo roskellyae, with deltran optimisation finding this character to be a synapomorphy of Wakaleo + Thylacoleo. Given that the upper tooth row of Priscileo roskellyae retains M4, it is probable that the lower tooth row retained M4. If this is found to be the case then the ambiguity will resolve in favour of the deltran optimisation.

Clade 2. Wakaleo + Thylacoleo

Unambiguous synapomorphies. Character 13 (1 to 2): P3 to M1 ratio greater than 1.5. Reversed in Wakaleo oldfieldi. Character 14 (0 to 1)*: central longitudinal blade of P3 longer than 50% of total tooth length. Character 21 (0 to 1)*: metaconule of M1 reduced to the point that it is barely developed or is lost altogether. Character 26 (1 to 2)*: an extremely strong anteroposterior gradient on the buccal side of M1 with a very tall anterior end. Character 33 (0 to 1)*: palatal ridges present.

Ambiguous synapomorphies under acctran optimisation. Character 28 (0 to 1)*: loss of metaconule on M2. Ambiguous due to missing data in Thylacoleo. This character state change is a synapomorphy of Wakaleo in delayed transformation.

Ambiguous synapomorphies under deltran optimisation. Character 2 (1 to 2): a P3to M1  ratio greater than 1.1. See discussion of this character above. Character 6 (0 to 1)*: trigonids of lower molars taller than the talonids. See discussion of this character above. Character 7 (0 to 1)*: talonid basins of lower molars distinctly narrower than the crown. See discussion of this character above.

Clade 3. Wakaleo

Unambiguous synapomorphies. Character 18 (0 to 1)*: presence of a lingual cuspule below the anterior cusp of P3. Character 20 (0 to 1)*: triangular occlusal outline of M1. Character 22 (0 to 1): loss of P1. The absence of the P1 in two of the outgroup taxa (Nimiokoala greystanesi and Namilamadeta albivenator) renders the optimisation of this character at the base of the tree ambiguous. Nevertheless this tooth is present basally in Thylacoleonidae and its loss can be unambiguously tied to Wakaleo within Thylacoleonidae. Character 25 (0 to 1)*: posterobuccal margin of M1 lengthened and overlapping M2 in lateral view. Character 27 (0 to 1): occlusal outline of M2 subtriangular to triangular.

Ambiguous synapomorphies under acctran optimisation. Character 3 (0 to 1): well-developed posterolingual crest on P3. Convergent in Thylacoleo hilli. Ambiguous due to missing data in W. alcootaensis. Deltran optimisation places this character state change on the branch supporting W. oldfieldi +W. vanderleueri. Character 23 (0 to 1): loss of P2. Ambiguous due to lack of data in W. oldfieldi and polymorphism in W. vanderleueri. Deltran optimisation interprets the loss of this tooth as an autapomorphy of W. alcootaensis, convergently acquired in some W. vanderleueri. Character 32 (0 to 1): recessed masseteric fossa. Reversed in W. vanderleueri. Deltran optimisation interprets this character state change as convergently evolved in W. oldfieldi and W. alcootaensis.

Ambiguous synapomorphies under deltran optimisation. Character 9 (0 to 1)*: trigonid of M2 much higher than its talonid. See discussion of this character above. Character 28 (0 to 1)*: loss of metaconule on M2. See discussion of this character above.

Clade 4. Wakaleo oldfieldi + Wakaleo vanderleurei

Unambiguous synapomorphies. Character 1 (0 to 1): presence of one alveolus between I1 and P3. Although it may seem likely that the single tooth present in this position in W. oldfieldi and W. vanderleueri is a primitive retention, it is interpreted here as a derived reacquisition due to the lack of teeth in this position in two of the outgroup taxa and the scarcity of data for basal thylacoleonids in this analysis. Character 34 (0 to 1)*: buccal margin of M2 much deeper than lingual margin.

Ambiguous synapomorphies under acctran optimisation. Character 2 (3 to 2): reversal to a P3: M1 less than 1.5. Character 11 (1 to 0): reversal to the presence of M3. See discussion of this character above.

Ambiguous synapomorphies under deltran optimisation. Character 3 (0 to 1): presence of a well-developed posterolingual crest on P3. Convergent in Thylacoleo hilli. See discussion of the character above.

Clade 5. Thylacoleo

Unambiguous synapomorphies. Character 1 (0 to 2): presence of two alveoli between I1 and P3. Although it may seem likely that the character state displayed by Thylacoleo is a primitive retention, it is interpreted here as a derived reacquisition due to the lack of teeth in this position in two of the outgroup taxa and the scarcity of data for basal thylacoleonids in this analysis. Character 4 (1 to 0): loss of anterobuccal crest on P3. Convergent in Pseudocheirus peregrinus. Character 14 (1 to 2)*: very long longitudinal blade of P3 more than 70% of total tooth length. Character 15 (1 to 2)*: loss of the posterior longitudinal blade of P3. Character 16 (1 to 0): loss of a mid-crown constriction of P3in occlusal view. Convergent in Nimiokoala greystanesi. Character 17 (1 to 0): loss of a posterobuccal crest on P3. A reversal of a character that evolved on the branch leading to Vombatimorphia. Character 19 (0 to 1)*: posterior width of P3 less than the anterior width.

Ambiguous synapomorphies under acctran optimisation. Character 10 (0 to 1)*: talonid of M2 lost. Ambiguous due to missing data for Thylacoleo hilli. Deltran interprets this chacter as a synapomorphy of Thylacoleo crassidentatus + Thylacoleo carnifex.

Additional Information and Declarations

Competing Interests

Author Contributions

Data Availability

The author declares he has no competing interests.

Adam M. Yates conceived and designed the experiments, performed the experiments, analyzed the data, contributed reagents/materials/analysis tools, wrote the paper, prepared figures and/or tables, reviewed drafts of the paper.

The following information was supplied regarding data availability:

The research generated a very small dataset in the Supplemental Information.

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
