# Peer review of "New craniodental remains of Wakaleo alcootaensis (Diprotodontia: Thylacoleonidae) a carnivorous marsupial from the late Miocene Alcoota Local Fauna of the Northern Territory, Australia"

_PeerJ, doi:10.7717/peerj.1408_

## Round 0.1 · original submission · Major Revisions

Dear author,

We have received three reviews for your manuscript “New craniodental remains of Wakaleo alcootaensis (Diprotodontia: Thylacoleonidae) a carnivorous marsupial from the late Miocene Alcoota Local Fauna of the Northern Territory, Australia”.

All the reviewers recommended the publication of this article requiring minor revisions, so please follow their suggestions (including those from the annotated PDF provided for Reviewer 1) or send a rebuttal if you disagree with what the reviewers (or the handling editor, see below) are requesting to be changed.

Please, take into account my own comments below, after submitting the revised version of the manuscript to PeerJ.

Concerning the status of the species of the genus Wakaleo, I would like to make some general comments that maybe will help you into find the best way to say what I presume you want to remark. Maybe you want to mean that Wakaleo alcootaensis can be differentiated from the supposed oldest species of the genus not only by its bigger size (cf. Archer and Rich, 1982), but also by other characters that you found are significant for species differentiation. That can be true, but you need to explain better your point, by considering the scarce information that there is available about this species. As general commentaries, you have to note some considerations: firstly, species do not define a lithostratigraphic unit, which means that one single species could be present in more than one formation or strata. So, when you say that the specimens that you are describing belong to the species Wakaleo alcootaensis for sure because they were found in the same strata that previous materials assigned to this taxon, is not a valid argument. Besides, in the Geological Setting section you describe very briefly the lithostratigraphy, and say almost nothing about the section that yielded the fossils. Indeed, the Waite Formation includes basal lacustrine greenish siltstone facies grading to reddish, coarser, fluvial deposits. I imagine you found the described materials at the fluvial facies, right? If not, you should provide some more stratigraphic details, as well as a taphonomic explanation for the fragmentary state of the W. alcootaensis specimens, considering the almost complete and well preserved condition of other materials collected from the Waite Formation. Please explain if it also occurs to the other species of the genus? Secondly, W. alcootaensis is placed as the most derived species of the genus, by stating that it has an enlarger premolar 4 and apparently, there is a cline to the progressive enlargement of this dentary element. However, taking a look at the previous works, I could see that W. oldfieldi appears to be the basalmost taxon of the series and W. vanderleueri is a little more derived. Nonetheless, both taxa are found associate at the same locality Riversleigh (cf. Gillespie et al., 2014), suggesting that the previous biostratigraphic usefulness proposed earlier for these taxa should be revised. Thus, providing more stratigraphic information for the new described materials, which you assigned to W. alcootaensis, surely will contribute for determine the existence of a hypothetical morphocline among the currently known species of the genus Wakaleo. But as reviewers recommended, you would have to include a phylogenetic analysis for to talk about morphoclines, as well as about parallel evolution of W. alcootaensis with the Thylacoleonidae.

In addition, I have some concerns with the diagnostic characters that you seem to have discovered for the better identification of W. alcootaensis, and whose are not properly remarked in your manuscript. As can be seen from the figures that you provided, W. alcootaensis lacks the M3 and also the small P2, (or ?P1) which are present in both the other species of the genus, right? This could be a good point to the easy recognition of W. alcootaensis, at least for specimens that represent mandibular remains. Moreover, if you see at the figure that I have attached to this letter, you will see that other arrangement of present teeth seems to be possible, particularly if you follow the labial alveolar margin of the dentary, which appears to mark the boundary of the preserved tooth positions (see the white arrows that I have added to your photography, which I also contrasted to see better the condition). Possible lingual constrictions of the teeth are not reflected at the lingual alveolar margin, but it is also much damaged. Intriguingly, when contrasted your figure 3 I could see a different catalog number on the lingual surface of the mandible than that you provided in the description. Thus, if the material that you are describing has previously pertained to other collection, you should indicate that in the manuscript. On the other hand, the mandibular fragment described by Woodburne (1967) and figured by you as Fig. 2, has two teeth that were originally identified as the M3 and M4. So, as you indicated in Fig. 2 that they are indeed M1 and M2, please, explain which are the criteria that you use to differentiate among molars and molars from premolars in Wakaleo species and in general, in thylacoleonids? I think that it is important to clarify this in the manuscript, considering the fragmentary nature of the specimens, which could lead to confusing interpretations of conditions that are diagnostic in the recognition of the species. Indeed, a brief description or a picture of the holotypic specimen (Archer and Rich, 1982) will be useful to show as a new figure. I think that the photos, which have not enough quality by the way, and the drawings (of lingual, labial and occlusal views) that you provided do not help to the resolution about the teeth that are present in NTM P4325. I have also added to the attached composite figure, for comparison, the occlusal view of a partial mandible of Wakaleo oldfieldi taken from Gillespie et al., 2014. If the hypothesis that you propose is valid, please, show it more clearly, with high quality figures (e.g. identifying the root holes that are corresponding to the preserved tooth positions, as you stated in the text). But take into account that maybe some molars could have three, instead two roots (if not, indicate a reference that shows the condition of just two roots present, which is that you are considering to identify the present teeth in NTM P4325).

As all the reviewers commented, I have also several concerns about the evolutionary and relationships section. You do not have support from a phylogenetic study for your hypothesis and as Reviewer 3 indicated, you have not considered alternative ones that are also possible. After to revising the issue about the present teeth in the mandible fragment or better, including a phylogenetic study, you have to improve this section or delete it.

In sum, I would like to see better images of the specimen that allows a more confident reconstruction of the dentition and so, a better diagnosis for W. alcootaensis and its relationships.

Despite all these concerns, I consider that the material that you are describing is very important to increase the knowledge of a poorly documented group of marsupial mammals. Thus, I will accept the manuscript under major revision, hoping that the new pictures will convince me about these other differences (beyond the larger size) that are relevant to distinguish Wakaleo alcootaensis from the congeneric W. oldfieldi and W. vanderleueri.

I look forward to see the revised manuscript soon.

With my best wishes,
Graciela Piñeiro

Reviewer 1 ·

Basic reporting

No comments

Experimental design

No comments

Validity of the findings

No comments

Additional comments

Review of: New craniodental remains of Wakaleo alcootaensis (Diprotodontia: Thylacoleonidae) a carnivorous marsupial from the late Miocene Alcoota Local Fauna of the Northern Territory, Australia.
By Adam Yates
This is a concise work that documents important new information about a very rare fossil species of marsupial.

Comments for the author
Numerous minor grammatical changes are noted in the pdf. However, a number of other issues listed below need to be addressed prior to recommendation for publication.
ABSTRACT
In the abstract and at the end of the introduction the author states that the work affirms that the species is "valid". This statement confers the idea that there was doubt about the identification and status of the species where there actually was none - its status as a species has never been in doubt and the statement should be removed - the species is substantially larger (by 30%) than its closest relative, W. vanderleueri, and also unquestionably younger, being late Miocene in age, in contrast to middle Miocene for W. vanderleueri.
INTRODUCTION
The introduction gives a brief description of the family, features of the genus Wakaleo, history of the discovery of the holotype specimen and a brief account of the recovery of the new material. The species is part of a morphocline and some mention of the other species in the morphocline, the nature of this morphocline and its position within the cline and relationship to the other species in the genus would add important and relevant background information.
line 11- The author states that marsupial lions are medium to large-bodied marsupials. This particular description fails to include the smallest described marsupial lion, Priscileo roskellyae, which is only cat size (estimated from its published dentition), and hence challenges the "medium" description. In addition, inclusion of a reference for estimated body size, for example by Wroe et al. (2003) would be useful. (Wroe, S. et al. An alternative to predicting body mass: the case of the Pleistocene marsupial lion, Paleobiology, 2003, 29 (3)).
line 18 - The reference Gillespie 2007, is missing from the references list and needs to be added.
line 24 - A summary of the Alcoota Local Fauna is given but lacks a reference - Murray and Megirian 1992.
line 25 - The size of W. alcootaensis is estimated as that of a female leopard - on what basis is this estimate made?
line 49 - as per comment above regarding the abstract - "status" statement is redundant and should be removed.
GEOLOGICAL SETTING
line 62 - age of the fauna is indicated as based on stage of evolution, but stage of evolution of what animals (molluscs?) - need to add what animal groups form the basis of the evolutionary studies.
line 63 - remove "almost certainly" - is redundant
line 64 remove "somewhere" - is redundant
METHODS
Terminology - A statement indicating what dental terminology is being followed should be included.
Additionally, anatomical terms should be standardised throughout the text and figures. Descriptions of marsupial dentitions usually employ the terms anterior/posterior, buccal (or labial)/lingual, occlusal, and dorsal/ventral. I have never encountered "mesial", and "distal" is usually used in relation to postcranial elements. These two terms are used in human dentistry and their use is far from common practice in palaeontology. For ease of comparison and understanding with other palaeontological descriptions it is recommended that these terms not be used.
RESULTS
lines 82 - 142 : Emended Diagnosis - diagnoses are normally quite concise, listing the main features that distinguish the taxon. In terms of the structure of the paper I think it would be improved by making the emended diagnosis more concise. This could be done simply stating the main diagnostic difference of each feature and moving much of the descriptive content to the appropriate areas of the description or discussion. This isn't essential but it would stop the reader becoming lost in descriptive comments. For example, for Larger size, the new material simply confirms a diagnostic feature of the original diagnosis of Archer and Rich, 1982, (that it is larger than other species of Wakaleo) and a simple statement to that affect is all that is required here (with reference to Table 1).
line 100 - change lateral to buccal.
line 107 - 112: I1 angle – the descriptions of the methods used to measure the I1 angle indicate that it is not consistent across species. The author indicates that the measurement for W. oldfieldi is made from the "base" of the I1 – I have assumed this is the ventral edge where the tooth exits the alveolus. This detail is significant because the different surfaces of the tooth will result in different angles - this is because the I1 tapers from root to tip and the angle the ventral surface of the tooth forms is different to the dorsal surface of the tooth. If the measurement has been made from the ventral surface this contrasts with the method used for W. alcootaensis which is measured from the dorsoposterior surface of the alveolus. I used the latter method on a dentary of W. vanderleueri that also lacked I1 (from the collection of the Queensland Museum), and obtained an I1angle of approximately 48 °, a value for that species that is much greater than the range for this species included in this paper. It is possible that the latter method results in an over-estimation of the angle of I1. Wakaleo alcootaensis may indeed have a more steeply-angled I1 but it would be more accurate in this instance to be more speculative and provide an estimated value rather than one that is so precise. Alternatively, it could be included as a "possible" diagnostic character. Additionally, to provide clarity, the author could mark on each dentary in Figure 5, the position where the arms of the angles were measured, rather than simply putting an angle to the right of each dentary.
DESCRIPTION - as mentioned earlier, the descriptions could be improved by using anterior instead of mesial and posterior instead of distal.
line 213 - reference is made to canine measurements "reported for W. vanderleueri" but a reference for this "report" is lacking.
lines 217 - 238 : Description of M2. There is potentially a significant problem with the description of this tooth stemming from an incorrect interpretation of its orientation which is evident in Figure 8. The caption for this figure indicates this tooth to be from the right side. If so, the interpretive drawings are incorrectly labelled. In figure D, the cusp indicating the metacone should be the protocone, the protocone should be the metacone and the orientation/anatomical direction symbol needs to be changed; in figure E (anterior view) the metacone and protocone need to be swapped. Following on from this interpretation, Figure C and F would therefore be a buccal view rather than a lingual view and thus the protocone should be the metacone. The incorrect orientation will require appropriate modifications to the description of the tooth. Another factor indicating the incorrect orientation is the measurements for the tooth (NTM P4328). The incorrect orientation probably explains the unusual dimensions for this tooth. As indicated in the text, usually Wakaleo M2s are broader than they are long.
DISCUSSION - the discussion lists the reasons why the new material is referred to W. alcootaensis. A more detailed comparison of the isolated M2 to the M2 and M3 of W. vanderleueri would have been interesting in terms of their morphoclinal evolution.
The paragraph concerning "Variation within Wakaleo alcootaensis" seems to add little value to the discussion because much of it concerns interspecific comparison with average values of W. vanderleueri that are not provided (see below). The only variation within the species is between the M2 of the holotype, for which there is no crown, and the isolated tooth which unfortunately has been oriented incorrectly and thus the measurements need to be reviewed and may not differ by the stated 15%.
line 263 - the statement regarding dental measurements as "on average" is inappropriate - average of what? From Table 1, of 7 different types of measurements, only two have more than one measure. Instead of "on average", and in relation to the measures in the table, a better phrase would be " in most cases". In addition, the author refers to the "mean values for W. vanderleueri" but there are no mean values provided in Table 1 . These should be added to the table to make comparisons easier if this section is going to be retained.
line 265 : again, "average" value is stated for W. vanderleueri but is not provided.
line 266-267: the statement regarding the size of the M2 needs review because of the mis-orientation of this tooth.
line 272 - statement regarding hypothesis of Thylacoleo not being ancestral to Wakaleo was put forward by earlier researchers and requires the appropriate reference (Clemens & Plane).
lines 273-275 ; source of derived characters requires appropriate reference (Gillespie).
REFERENCES
Gillespie 2007 - is missing
FIGURES: in the figure captions, views should be consistent throughout and not change from one figure to another, i.e. all buccal, rather than lateral in one and buccal in the next.
Figure 1 - Needs the addition of a small map of Australia showing location of Alcoota within the Australian continent.
Figure 2 - change B, from dorsal to occlusal
Figure 3 - change A, from lateral to buccal, B, from medial to lingual.
Figure 4 - change A from lateral to buccal, B, from medial to lingual. Labels A, B and C are missing from figure.
Figure 5 caption - tooth numbers need to be subscript.
The photographs in Figures 2, 3 and 7 are too pale and need to be darker. It is difficult to see any depth - thus it is difficult to see the important diagnostic feature of the "deeply recessed masseteric fossa".
Figure 8 is not correctly labelled - the orientation is incorrect and thus, so are most of the labels.
Additionally, the photographs in Figure 8, could be improved – it is difficult to discern any relief. Normally figures of tooth crowns are presented as stereo photographs and Figure A should be presented as such.

Annotated reviews are not available for download in order to protect the identity of reviewers who chose to remain anonymous.

·

Basic reporting

Introduction: lines 23-30 many sentences are really short, making the paragraph odd to read, please change the writing.
lines 37-46 are unnecessarily long. Be more precise with the idea.
Systematic Paleontology. Please cite the figures in order. Fig. 6 is mentioned before Fig. 5
Line 82: Change the title Emmended Diagnosis to Description or Morphological Comparison or other title. What the author is doing is not a diagnosis in the nomenclatural sense.
Define the abbreviations used for the teeth, not all the readers might be familiar with them.
Figures need to be corrected. Figs 3 and 7 have different tones of black in them, and contrast of photographs of fig 2, 3, 7, 8 need to be improved

Experimental design

Descriptions are well performed. It is a good taxonomic paper.

Validity of the findings

My only concern is with the section Parallel evolution within Thylacoleonidae. The author needs to perform a phylogenetic analysis in order to test the hypothesis of this section. You cannot talk of derived or ancestral characters without a phylogenetic context. I believe that this section must be ommited, or at least clearly state that the hypothesis need to have a phylogenetic background in order to be tested.

·

Basic reporting

The manuscript is well organised and presented. Sufficient background is given on the topic. The structure conforms with the PeerJ template. Figures are relevant and well illustrated. A methods section is missing, which could easily be added, to describe terminology used, how specimens were measured, etc.

Experimental design

The submission is original and represents important new material.
Again, methods are missing and could be added to improve the manuscript.

Validity of the findings

The interpretation of the specimens is sound and valid, but the conclusions about the evolution in Thylacoleonids does not explore all possible outcomes. The author concludes in the discussion that Wakaleo couldn't be the ancestral to Thylacoleo because of the lack of P1/p1. Though, the author lists four characters shared between Thylacoleo and Wakaleo alcootaensis. It seems the most parsimonious outcome is that a character reversal occurred, and that P1/p1 remerged. There are plenty of example of character reversal in the literature, and this could be easily tested for Thylacoleonids by simply running a phylogenetic analysis. Though, this is not in the scope of this paper, but the alternative most parsimonious hypothesis should be mentioned here.

Additional comments

Overall, it's a very good paper and should get published.

---

## Round 0.2 · Minor Revisions

Dear author,

I am very pleased to read this improved version of your manuscript entitled “New craniodental remains of Wakaleo alcootaensis (Diprotodontia: Thylacoleonidae) a carnivorous marsupial from the late Miocene Alcoota Local Fauna of the Northern Territory, Australia”.
The manuscript was indeed modified to produce a strengthened paper. The figures look fine now and I liked very much that you have decided to include additional ones that resulted very informative and provided fundamental evidence for validate some points of your original interpretations that appeared to show some weakness. I accept all the explanations provided in the rebuttal letter regarding the reasons that lead you to retain your original interpretations, despite the suggestions of revision and changes recommended by reviewers and also from myself. Nevertheless, I think that those comments and requests were useful for you to complement and reinforce previous argumentations and most of all, left clear that this is a well-supported, valuable contribution to the knowledge of the Thylacoleonidae evolution.

I am attaching an annotated version of the last pdf file that you submitted where you will find just a single observation, asking for a revision of the sentence corresponding to the Emended Diagnosis in line 182. After fixing this minor matter, I consider that the manuscript can be accepted for publication in PeerJ.

Best regards,

Graciela Piñeiro

---

## Round 0.3 · accepted · Accept

Thank you for the swift revision of your manuscript. I am delighted now to accept it for publication at PeerJ. I hope you consider publishing with PeerJ in the future.
Best regards,
Graciela Piñeiro